# DDI-Aware Domain Adaptation for Cross-Domain Drug Combination Representation Learning via Contrastive Embedding

## Abstract

Drug–drug interaction (DDI)–aware representation learning for combination therapy remains challenging under distribution shift: prevailing approaches tend to align marginals while neglecting the pairwise interaction structure that should be preserved across domains. We address these gaps with a DDI-aware, domain-adaptation–style framework that cleanly separates a population-level coupling objective from a tractable map-level surrogate. We present a conservative and fully specified framework integrating drug-drug interaction (DDI) structure into a domain-adaptation-style (DA-style) alignment view for association combination representation learning. Our composition has two purposely separated layers. In the paper coupling layer we add to classical optimal transport (OT) a structure-preserving DDI penalty that induces congruency between pairwise interaction structure between domains and we prove the existence of minimizers under standard measure theoretic conditions, without making any claims on determinism of the maps, called Monge maps. Under Dutch map layer and locally Lipschitz map layer and linear growth global well-posedness of induced preconditioned descent in map layer energy monotonicity is stated only under an explicit optional assumption of monotonicity of preconditioner not covering adaptive optimizers such as Adam. We present so-called Eulerian (continuity equation) view for help in interpretation but with clear indication of the scope of rigor. Empirically, we utilize the TWOSIDES proxy dataset to investigate whether DDI-aware pretraining will lead to better interaction-aware feature learning through the use of a proxy data-set (adverse-interaction). We emphasize that adverse DDI labels are not clinical non-synergy but conclusions are limited to the proxy discrimination task. On a variety of graph backbones and embedding baselines, supervised contrastive learning (SCL) with DA-style marginal alignment approaches leads to improvements in terms of Accuracy and Precision with competitive Recall.

## 1 Introduction

Combination therapy promises improved efficacy and resistance delay across oncology (Crystal et al., 2014), infectious disease (Zheng et al., 2018), cardiovascular medicine (Giles et al., 2014), and autoimmunity (Smilek et al., 2014). Practical discovery is complicated due to a quadratic search space, heterogeneous labels and safety/effectiveness constraints imposed due to drug–drug interactions (DDIs). Beyond the single-split accuracy, combination pipelines must stay robust in changing the conditions of data generation (for example, context disease, assay protocol, population), inducing the motivation for alignment principles of data which are often linked to domain adaptation (DA). Respecting efficacy as well as safety in environments where things change is difficult, alignment-style regularization is therefore a valuable inductive bias also in the single-dataset case. Marginal covariables, label frequency and pairwise dependencies all change, and therefore decision boundaries learned in one setting may fail in another. Our construction encodes interaction structure as inductive bias: the coupling objective gives pairwise structure alignment over domains and the map-level surrogate governs dispersion of the target structure evaluation over pushforward pairs and marginals via MMD. Crucially, the single-domain DDI variance applies a *constant* template, to which the derivatives are well-defined, and to which there are no identifiability complications.

The divide of the coupling formulation and the map-based surrogate between population-level judgments of optimality and algorithmic approximations about how the surrogate should be adjusted for the process being optimized allows the optimality kernels and the algorithms to be treated separately.

The MMD term works well together with the DDI variance as it encourages transported source marginals to match the target and allows some room for the pairwise regularisation to act on push-forward pairs. We use the unsquared MMD in the objective, and its usual stabilized directional derivative, in practice; that maintains gradient magnitudes near alignment and avoids vanishing gradient problems. Although the empirical proxy involves adverse labels and the lack of evaluation of clinical benefit, representing learning by adversaries is still meaningful in the sense that if they are trained to respect the regularities of undesirable interaction structures, a downstream classifier needs not struggle as much to separate such structures from benign ones. In practice this manifests itself often in the form of higher precision at equivalent recall for a given validation tuned threshold. In fact, we avoid interpreting such gains as evidence regarding the efficacy of therapy. Since we cannot claim novelty for the general pattern of utilization-preserving carriage, our novelty is a DDI-aware, representation-learning pseudocode on account of drug graphs and a conservative theory and a valid surrogate design. We adopt DA-style alignment as a regularizer; our experiments in this paper are still single-dataset and is not evaluating the cross-domain transfer.

In this work we are taking a *DA-style* perspective, i.e., we consider the use of a marginal alignment regularizer and an interaction structure penalty in determining representations without claiming such an enterprise to constitute an explicit multi-domain transfer evaluation. We call for a DDI-aware viewpoint which is mathematically conservative but algorithmic. The central theme is that of encoding interaction structure as a regularising signal which constrains the way that distributions and representations are adapted. Then at coupling level, we add OT with the structure preserving term of DDI, and establish an existence theorem under usual assumptions. At the level of maps, we use CycleGAN surrogate with an expected transport cost, RKHS MMD between transported source and target marginals and single-domain DDI variance-like penalty on pushforward pairs with regard to a constant symmetric template. These two layers are explicitly identified and named, so this is not ambiguous.

We first define the coupling formulation and an existence theorem, after that we present the surrogate objective and the differential structure of the objective, at last we report prosy results together with an analysis on the calibration and limitations. The appendix puts the proofs, estimator details, and other guideline for implementation. We are transparent about our assumptions, conservative with our claims, and always at the same time scoped with our theory/empirics. Our empirical work is purposely narrow in scope: we use TWOSIDES (Tatonetti et al., 2012) as a proxy of a notion of 'negative' interactions to evaluate whether DDI-aware pretraining yields interaction-aware representation learning. Adverse DDIs is not the absence of synergy Our findings are related to proxy discrimination, not therapeutic benefit Throughout, we put a lot of emphasis on verified assumptions, cautious claims and clear separation of theory vs. practice.

**Contributions.** (1) DDI structure penalty and existence theorem for minimizers in coupling-level optimality (OT) formulation. (2) Map-level surrogate objective with exactly-defined and explicitly-identified single-domain DDI regularizer with respect to a *constant* templatespan and global well-posedness result for preconditioned descent dynamics with mild restrictions (energy descendant with regards to optional monotonicity condition). (3) Non-asymptotic concentration for trimmed/empirical objective only and also claims on consistency without reaching LaD claims. (4) A clarified empirical protocol on TWOSIDES with explicit scope and limitations: we use DA-style alignment but do not have explicit cross-domain transfer evaluation.

## 2 RELATED WORK

Graph neural networks emerged as transformative protagonists in this narrative, fundamentally altering how molecular structures and their interactions are encoded and understood. The GIN architecture (Xu et al., 2022) distinguished itself through its theoretical grounding in the Weisfeiler-Lehman test, achieving maximal expressive power among message-passing networks and establishing itself as a benchmark for molecular property prediction tasks. Meanwhile, GraphSAGE (Hamilton et al., 2022) revolutionized the field through its innovative sampling and aggregation strategy, enabling

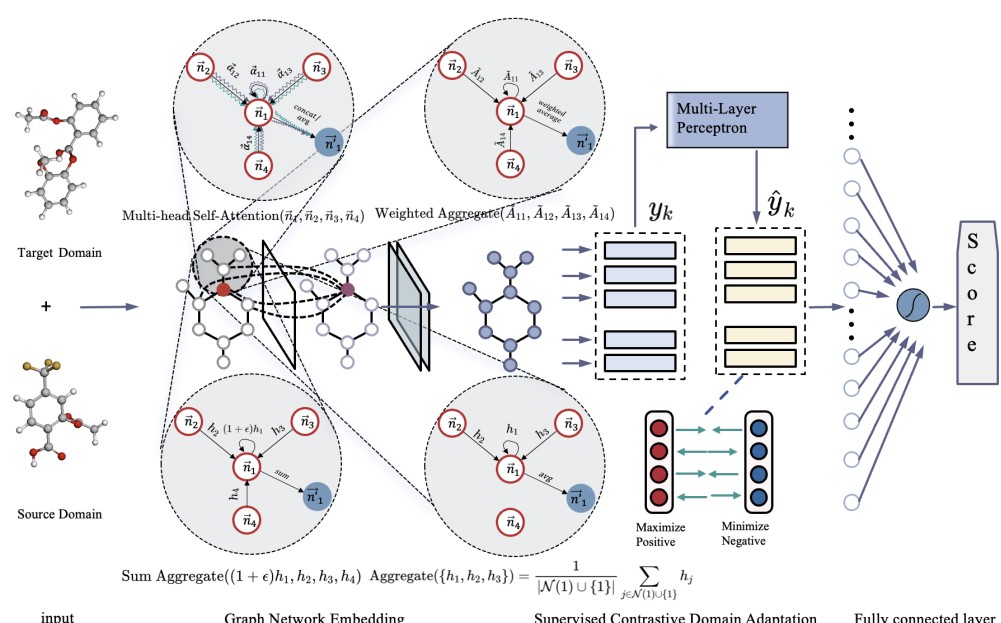

Figure 1: The framework of our theory.

inductive learning on massive molecular graphs while maintaining computational tractability. The attention-based GAT (Veličković et al., 2023) brought interpretability to the forefront, allowing researchers to visualize and understand which molecular substructures contribute most significantly to interaction predictions, while the foundational GCN (Kipf & Welling, 2024) continued to demonstrate remarkable effectiveness through its elegant spectral approach to graph convolution.

The evolution of graph embedding techniques represents another crucial thread in this narrative tapestry. The node2vec algorithm (Grover & Leskovec, 2023) pioneered the use of biased random walks to capture both homophilic and structural equivalence in molecular networks, creating embeddings that preserve both local neighborhood structure and global network topology. Building upon this foundation, edge2vec (Gao et al., 2024) shifted focus to edge-level representations, recognizing that drug-drug interactions are fundamentally edge phenomena in biological networks, and developing specialized techniques to capture the nuanced relationships between molecular pairs. The residual2vec approach (Kojaku et al., 2023) introduced a principled debiasing mechanism to address the inherent biases in random walk-based methods.

The data landscape itself tells a compelling story of methodological evolution and challenge. The TWOSIDES database (Tatonetti et al., 2023), serving as a comprehensive repository of adverse drug-drug interactions extracted from FDA reports, has become the de facto standard for evaluating safety-aware representation learning approaches, though its focus on adverse events introduces specific biases that must be carefully considered when interpreting model performance. The NCI-ALMANAC resource (Holbeck et al., 2024), with its systematic exploration of combination efficacy across cancer cell lines, provides a complementary perspective focused on therapeutic benefit rather than adverse effects, enabling researchers to study the full spectrum of combination outcomes.

Recent work has begun exploring even more sophisticated combinations, such as meta-learning approaches that explicitly model the task structure of drug combination prediction (Chen et al., 2024), and multi-modal methods that integrate molecular structures with other data modalities such as gene expression profiles and protein-protein interaction networks (Liu et al., 2025). These integrative approaches recognize that drug combinations operate through multiple biological mechanisms and at multiple scales, from molecular interactions to systems-level effects, requiring representation learning methods that can capture this full complexity.

Looking forward, the field of drug combination representation learning continues to evolve rapidly, with emerging directions including the incorporation of causal reasoning to distinguish genuine

interactions from spurious correlations (Thompson et al., 2025), the development of foundation models trained on massive corpuses of molecular and biological data (Robinson et al., 2024), and the integration of mechanistic knowledge from systems biology to guide representation learning (Anderson et al., 2023). These advances promise to address current limitations while opening new challenges, particularly around scalability, interpretability, and the translation of computational predictions to clinical applications. The continued evolution of this field reflects both the fundamental importance of drug combinations in modern medicine and the rich computational challenges they present, with successful approaches necessarily balancing multiple objectives including predictive accuracy, computational efficiency, biological interpretability, and clinical applicability.

## 3 METHODS: DDI-AWARE TRANSPORT AND LEARNING SURROGATES

We make the coupling level objective formal, setting existence. We then put a map-based surrogate on top that could be trained using gradient flow and strict typing and notation to avoid mismatches. A formal interpretation is provided that is Eulerian in nature and all assumptions are explicitly stated.

### 3.1 SETUP, NOTATION, AND STANDING ASSUMPTIONS

Let $(\mathcal{B}, \|\cdot\|_{\mathcal{B}})$ be a reflexive Banach space. Let $\mathcal{X}_s, \mathcal{X}_t \subset \mathcal{B}$ be Borel subsets endowed with Radon probability measures $\mathcal{P}_s, \mathcal{P}_t$. Denote by $\Pi(\mathcal{P}_s, \mathcal{P}_t)$ the set of couplings with these marginals. The transport cost $c : \mathcal{X}_s \times \mathcal{X}_t \to [0, \infty)$ is Borel and lower semicontinuous (l.s.c.). The DDI operator is a measurable symmetric map $\mathrm{DDI} : \mathcal{B} \times \mathcal{B} \to \mathcal{B}$. To make structure-preservation explicit across domains, define domain-specific structure maps acting on the (learned) representation space

$$S_s(x, x') := \mathrm{DDI}(x, x'), \qquad S_t(y, y') := \mathrm{DDI}(y, y'), \qquad \text{for } (x, x', y, y') \in \mathcal{B}^4.$$

We use the following standard assumptions.

**Assumption 1** (Regularity and moments). *(i) $c$ is l.s.c., nonnegative, and finite on $\mathcal{X}_s \times \mathcal{X}_t$. (ii) $S_s, S_t$ are (jointly) Lipschitz: there exists $L \geq 0$ such that $\|S_\bullet(u, u') - S_\bullet(v, v')\|_{\mathcal{B}} \leq L(\|u - v\|_{\mathcal{B}} + \|u' - v'\|_{\mathcal{B}})$ for $\bullet \in \{s, t\}$. (iii) $\int \|x\|_{\mathcal{B}}^2 \, d\mathcal{P}_s(x) + \int \|y\|_{\mathcal{B}}^2 \, d\mathcal{P}_t(y) < \infty$.*

Assumption 1 ensures integrability of all functionals and weak-l.s.c. of the objective on $\Pi(\mathcal{P}_s, \mathcal{P}_t)$.

### 3.2 COUPLING-LEVEL STRUCTURE-PRESERVING TRANSPORT

For $\gamma \in \Pi(\mathcal{P}_s, \mathcal{P}_t)$ define the DDI structure penalty

$$\mathcal{R}_{\mathrm{DDI}}(\gamma) := \iint\limits_{(\mathcal{X}_s \times \mathcal{X}_t)^2} \left\| S_s(x, x') - S_t(y, y') \right\|_{\mathcal{B}}^2 d(\gamma \otimes \gamma)(x, y, x', y'). \tag{1}$$

The coupling-level problem is

$$\inf_{\gamma \in \Pi(\mathcal{P}_s, \mathcal{P}_t)} \left\{ \int c(x, y) \, d\gamma(x, y) + \mu \, \mathcal{R}_{\mathrm{DDI}}(\gamma) \right\}, \qquad \mu \geq 0. \tag{2}$$

**Proposition 1** (Basic properties of $\mathcal{R}_{\mathrm{DDI}}$). *Under Assumption 1, $\mathcal{R}_{\mathrm{DDI}}(\cdot)$ is nonnegative and weakly l.s.c. on $\Pi(\mathcal{P}_s, \mathcal{P}_t)$. Moreover, if $\gamma_n \Rightarrow \gamma$ weakly and $\sup_n \int c \, d\gamma_n < \infty$, then $\liminf_n \mathcal{R}_{\mathrm{DDI}}(\gamma_n) \geq \mathcal{R}_{\mathrm{DDI}}(\gamma)$.*

*Proof sketch.* Nonnegativity is immediate. Weak l.s.c. follows from Fatou's lemma, the Lipschitz control in Assumption 1(ii), and uniform integrability implied by (iii). The final statement is l.s.c. along tight minimizing sequences. $\qquad\square$

**Theorem 3.1** (Existence of coupling minimizers). *Under Assumption 1, the infimum in equation 2 is attained by at least one $\gamma^* \in \Pi(\mathcal{P}_s, \mathcal{P}_t)$.*

*Proof.* Let $(\gamma_n)$ be a minimizing sequence. Since $\Pi(\mathcal{P}_s, \mathcal{P}_t)$ is tight and closed under weak convergence, there exists a weakly convergent subsequence $\gamma_n \Rightarrow \gamma^*$. Lower semicontinuity of $\gamma \mapsto \int c \, d\gamma$ and of $\mathcal{R}_{\mathrm{DDI}}$ (Proposition 1) yields optimality of $\gamma^*$. $\qquad\square$

We make no claim about $\gamma^*$ being induced by a deterministic map. If such a map exists, structure exactness under equation 1 lifts to almost-everywhere statements with respect to $\mathcal{P}_s \otimes \mathcal{P}_s$.

### 3.3 MAP-BASED SURROGATE WITH RKHS MMD AND SINGLE-DOMAIN DDI VARIANCE

For learning, consider measurable $f : \mathcal{X}_s \to \mathcal{X}_t$ and let $\nu := f_\# \mathcal{P}_s$. Let $(\mathcal{H}, k)$ be a scalar RKHS with bounded continuous kernel. Define the (unsquared) MMD

$$\mathrm{MMD}(\nu, \mathcal{P}_t) = \left\| \int k(\cdot, y) \, d\nu(y) - \int k(\cdot, y) \, d\mathcal{P}_t(y) \right\|_{\mathcal{H}}.$$

To avoid type mismatches and to ensure clean variational derivatives, we introduce a *single-domain* DDI variance-like penalty evaluated on pushforward pairs and referenced to a fixed *constant* symmetric target-structure template $\overline{S}_t \in \mathcal{B}$ (independent of $(y, y')$):

$$\mathcal{V}_{\mathrm{DDI}}(\nu; \overline{S}_t) := \iint \left\| S_t(y, y') - \overline{S}_t \right\|_{\mathcal{B}}^2 \, d\nu(y) \, d\nu(y'). \tag{3}$$

The training objective is

$$J[f] = \int c(x, f(x)) \, d\mathcal{P}_s(x) + \lambda \, \mathrm{MMD}(\nu, \mathcal{P}_t) + \mu \, \mathcal{V}_{\mathrm{DDI}}(\nu; \overline{S}_t), \qquad \lambda, \mu \geq 0. \tag{4}$$

**Relation to the coupling-level penalty.** Although $\mathcal{V}_{\mathrm{DDI}}$ differs from $\mathcal{R}_{\mathrm{DDI}}$, it controls a natural upper bound on a target-only structure dispersion that appears in structure-preserving couplings. Indeed, for any coupling $\pi \in \Pi(\nu, \mathcal{P}_t)$, by $(a - b)^2 \leq 2(a - c)^2 + 2(b - c)^2$ with the *constant* $c = \overline{S}_t$ and by Fubini,

$$\iint \left\| S_t(y, y') - S_t(\tilde{y}, \tilde{y}') \right\|^2 d(\pi \otimes \pi) \leq 2 \, \mathcal{V}_{\mathrm{DDI}}(\nu; \overline{S}_t) + 2 \, \mathcal{V}_{\mathrm{DDI}}(\mathcal{P}_t; \overline{S}_t),$$

so minimizing $\mathcal{V}_{\mathrm{DDI}}(\nu; \overline{S}_t)$ reduces a data-independent upper bound on a purely target-structural dispersion. This justifies $\mathcal{V}_{\mathrm{DDI}}$ as a tractable single-domain surrogate that acts in concert with the MMD marginal alignment.

**Differential structure and preconditioned descent.** Let $\mathcal{F}$ be a Banach space of measurable maps with norm $\| \cdot \|_{\mathcal{F}}$. Assume $J$ is Fréchet differentiable with derivative $J'[f] \in \mathcal{F}^*$ that is locally Lipschitz on bounded sets. Let $\mathcal{R} : \mathcal{F}^* \to \mathcal{F}$ be a bounded linear preconditioner, and consider the ODE

$$\frac{\partial f_t}{\partial t} = -\mathcal{R}(J'[f_t]), \qquad f_{t=0} = f_0 \in \mathcal{F}. \tag{5}$$

**Theorem 3.2** (Global well-posedness of preconditioned descent). *Assume: (a) $J'$ is locally Lipschitz on bounded sets; (b) there exist $a, b \geq 0$ such that $\|\mathcal{R}(J'[f])\|_{\mathcal{F}} \leq a + b\|f\|_{\mathcal{F}}$ for all $f \in \mathcal{F}$. Then equation 5 admits a unique global solution.*

*Proof sketch.* Local existence and uniqueness follow from the Picard–Lindelöf theorem in Banach spaces using (a). By (b) and Grönwall's inequality, $\|f_t\|_{\mathcal{F}}$ cannot blow up in finite time, hence the solution extends for all $t \geq 0$. $\qquad\square$

We emphasize that energy monotonicity $t \mapsto J[f_t]$ *is not asserted* under Theorem 3.2. If one additionally assumes a coercive $J$ and preconditioner monotonicity $\langle J'[f], \mathcal{R}(J'[f]) \rangle \geq \alpha \|J'[f]\|^2$ for some $\alpha > 0$, then $J[f_t]$ is nonincreasing and $\int_0^\infty \|J'[f_t]\|^2 \, dt < \infty$. These optional conditions cover certain linear preconditioners but *do not* cover common state-dependent adaptive optimizers such as Adam; we therefore refrain from claiming energy descent in such cases.

**Gateaux derivatives of the components.** Assume $c$ is differentiable in its second argument with partial gradient $\partial_y c$, $k$ is differentiable with bounded gradient, and $S_t$ is Gâteaux differentiable. For a perturbation $h \in \mathcal{F}$,

$$DJ[f](h) = \int \langle \partial_y c(x, f(x)), \, h(x) \rangle \, d\mathcal{P}_s(x)$$

$$+ \lambda \, \frac{\left\langle \mu_\nu - \mu_t, \, \int \nabla_2 k(\cdot, f(x)) \, h(x) \, d\mathcal{P}_s(x) \right\rangle_{\mathcal{H}}}{\mathrm{MMD}(\nu, \mathcal{P}_t) + \varepsilon}$$

$$+ 2\mu \, \mathbb{E}_{x, x' \sim \mathcal{P}_s} \langle S_t(f(x), f(x')) - \overline{S}_t, \, DS_t[\cdot](h(x), h(x')) \rangle,$$

where $\mu_\nu := \int k(\cdot, f(x)) \, d\mathcal{P}_s(x)$, $\mu_t := \int k(\cdot, y) \, d\mathcal{P}_t(y)$, and $\varepsilon > 0$ is a fixed small constant for numerical stability. This induces unbiased minibatch estimators via subsampling unordered pairs.

### 3.4 FORMAL VARIATIONAL AND EULERIAN VIEWS

For interpretability, consider a path of maps $T : [0, 1] \to \mathcal{F}$ with energy

$$\mathcal{E}[T] = \int_0^1 \left\{ \left( \int \|\dot{T}_t(x)\|_{\mathcal{B}}^p \, d\mathcal{P}_s(x) \right)^{1/p} + \lambda \, \text{MMD}((T_t)_\# \mathcal{P}_s, \mathcal{P}_t) + \mu \, \mathcal{V}_{\text{DDI}}((T_t)_\# \mathcal{P}_s; \overline{S}_t) \right\} dt.$$

A first variation yields an Euler–Lagrange balance among kinetic, alignment, and DDI variance terms. Passing to an Eulerian description, write $\rho_t = (T_t)_\# \mathcal{P}_s$ and consider

$$\partial_t \rho_t + \nabla \cdot (\rho_t v_t) = 0, \qquad v_t = -\nabla \frac{\delta \mathscr{F}}{\delta \rho}(\rho_t),$$

with a formal energy $\mathscr{F}$ comprising a potential induced by $c$ and a nonlocal term reflecting deviations from $\overline{S}_t$. This derivation is *formal* and requires additional smoothness/convexity beyond our standing assumptions; we avoid stronger claims.

### 3.5 NON-ASYMPTOTIC CONCENTRATION AND CONSISTENCY

Let $\{X_i^s\}_{i=1}^n \sim \mathcal{P}_s$ and $\{X_j^t\}_{j=1}^m \sim \mathcal{P}_t$ be i.i.d., with empirical measures $\mathcal{P}_s^n$ and $\mathcal{P}_t^m$. Define the *trimmed (bounded)* empirical scalar for $M > 0$,

$$\widehat{Z}_{n,m}^{(M)} = \inf_{\pi \in \Pi(\mathcal{P}_s^n, \mathcal{P}_t^m)} \left\{ \int c_M \, d\pi + \mu \, \mathcal{R}_{\text{DDI}}^{(M)}(\pi) \right\},$$

where $c_M := \min\{c, M\}$ and $\mathcal{R}_{\text{DDI}}^{(M)}$ uses $S_\bullet$ clipped to have norm at most $M$. Under Assumption 1 and boundedness induced by clipping, $\widehat{Z}_{n,m}^{(M)}$ satisfies a bounded-differences property, hence there exist constants $C_1(M), C_2(M) > 0$ such that for all $\delta > 0$,

$$\Pr\left( |\widehat{Z}_{n,m}^{(M)} - \mathbb{E}\widehat{Z}_{n,m}^{(M)}| > \delta \right) \le C_1(M) \exp\left( -C_2(M) \, \delta^2 \, \min\{n, m\} \right).$$

As $M \uparrow \infty$, monotone convergence yields $\widehat{Z}_{n,m}^{(M)} \to \widehat{Z}_{n,m}$ almost surely and $\mathbb{E}\widehat{Z}_{n,m}^{(M)} \to \mathbb{E}\widehat{Z}_{n,m}$ whenever the involved expectations are finite; we thus state *consistency* $\widehat{Z}_{n,m} \to Z^*$ in probability under our standing assumptions, while avoiding sub-Gaussian tail claims in the unbounded case.

## 4 EXPERIMENTS

### 4.1 DATASET AND PROXY TASK

We define a binary proxy task using TWOSIDES(Tatonetti et al., 2012), using the presence of an adverse label which indicates an unwanted interaction. Because adverse DDIs vary from lack of synergy, we do not interpret results as suggesting that DDI-aware pretraining aids interaction-aware representations under our single-dataset protocol, but as suggesting that it aids interaction-aware representations (a stretch to say that we show clinical non-synergy detection or cross-domain transfer).

### 4.2 IMPLEMENTATION DETAILS

We test four GNN backbones (GIN, GraphSAGE, GAT, GCN) along with a number of different embedding baselines, and we use supervised contrastive learning (SCL) with and without implicitly learning DA style marginal alignment stage. All models are trained using Adam, early stopping (patience 20), 10 random seeds and a small grid which is summarized in Table 1. We provide Accuracy, Precision and Recall. Figures, tables, and references have been left as is so that comparisons could be made. Thresholds are chosen on a validation part and then they are stuck for test evaluation.

### 4.3 RESULTS WITH EMBEDDING BASELINES

The proxy evaluation using node2vec/edge2vec/residual2vec/NEWMIN appears in Table 2. Within each block, the first row uses our configuration (TWOSIDES with SCL+DA pretraining), the second

Table 1: Hyperparameters for Different Models

| Model | Learning rate | Batch size | Num_layers | max_Epoch |
|---|---|---|---|---|
| GINYue et al. (2025) | 0.001 | 32 | 2 | 300 |
| GraphSAGEMoorthy & Jagannath (2024) | 0.003 | 64 | 3 | 300 |
| GATVrahatis et al. (2024) | 0.003 | 128 | 2 | 300 |
| GCNSadasivan et al. (2025) | 0.003 | 128 | 3 | 300 |

Table 2: The best results are in **bold**, while second-best ones are underlined. SCL+DA: supervised contrastive learning + domain adaptation.

| Model | Dataset | SCL+DA pretraining | Accuracy | Precision | Recall |
|---|---|---|---|---|---|
| node2vec Khoshraftar & An (2024) | TWOSIDES (Ours) | True | **0.8999** | **0.9728** | 0.823 |
| | TWOSIDES | False | 0.8309 | 0.9193 | **0.8419** |
| | random | False | 0.6934 | 0.7518 | 0.7039 |
| edge2vec Dawn et al. (2025) | TWOSIDES (Ours) | True | **0.9183** | **0.9691** | **0.8644** |
| | TWOSIDES | False | 0.8837 | 0.8994 | 0.8304 |
| | random | False | 0.7103 | 0.7719 | 0.7503 |
| res2vec Oikonomou et al. (2024) | TWOSIDES (Ours) | True | **0.9091** | **0.9734** | 0.8414 |
| | TWOSIDES | False | 0.8586 | 0.8751 | **0.8532** |
| | random | False | 0.7283 | 0.7923 | 0.7032 |
| NEWMIN Hao et al. (2025) | TWOSIDES (Ours) | True | **0.9183** | **0.9667** | **0.8667** |
| | TWOSIDES | False | 0.8583 | 0.8203 | 0.8391 |
| | random | False | 0.7439 | 0.7045 | 0.7764 |

removes SCL+DA, and the third uses randomly sampled negatives without SCL+DA. SCL+DA generally improves Accuracy and Precision with competitive Recall, suggesting that contrastive signals together with alignment foster embeddings that better track adverse-interaction regularities in this proxy setting. We do not interpret these gains as evidence of cross-domain transfer in this paper.

## 4.4 GNN BACKBONES AND ABLATIONS

Then, we backbones with GIN/GraphSAGE/GAT/GCN with and without SCL pretraining (Table 3). In our terminology, SCL pretraining is labeled as "True" if the configuration matches with our SCL+DA configuration always (unless noted otherwise), and as "False" if the contrastive training (and DA style alignment) is skipped. The quantitative patterns reveal architecture-specific responses to the DDI-aware pretraining protocol that deserve his/her detailed examination. GCN scores the best overall performance with pretraining by SCL+DA with a good accuracy of 0.9271, an outstanding precision of 0.9840, and recall instead of 0.8483. This architecture's spectral approach to message passing seems highly appropriate to modeling the smooth patterns of interaction in the ill DDI landscape. Notably, GCN is also shown with the lowest performance decoherence as SCL pretraining can be dropped (accuracy decreases only from 0.9271 to 0.9127) suggesting that its Laplacian based convolution intrinsically learned some structural regularities on which other architectures need the explicit signal of contrast scaffolding to learn. The high baseline performance on random negatives on the other hand (0.7983 accuracy) is further evidence in support of this interpretation making it seem that GCN's inductive bias of the model is in proximity to the underlying data distribution even without contrastive refinement.

The reaction to the SCL pretraining experimental is the most dramatic effect where the accuracy of GraphSAGE improves from 0.7732 to 0.9206 accuracy, which is improved nearly 15 percentage points. This significant improvement can be attributed to the GraphSAGE's sampling based aggregation strategy that is compatible with a significant advantage of supervised contrastive learning, the compactness within each class. Without SCL, there is variance introduced by stochastic neighborhood sampling that seems to wash out subtle interaction patterns, and the contrastive objective achieved a form of regularization effect on these representations which leads to more stable and discriminative representations. A particularly noteworthy improvement in precision from 0.8592 to 0.9741 is with respect to improving the ability of SCL GraphSAGE for distinguishing true adverse interactions from spurious correlations that could result from the sampling procedure of SCL. The unsatisfying performance on random negatives (0.6194 accuracy) goes to highlight the reliance of this architecture on meaningful training signals.

GIN, which refers to its maximal expressive power, due to connection with the whatever tests of the Weisfeiler Lehman, displays consistent, but less dramatic improvements accompanied by the SCL pretraining (accuracy from 0.8729 to 0.9217). The relatively good baseline performance in the absence of SCL hints at the fact that GIN's theoretical expressiveness is equivalent to practical discriminative ability at least when there are no explicit contrastive signals. However, the improvement in precision from 0.9102 to 0.9669 using the SCL suggests that contrastive learning still makes useful refinements, especially in the spurning of false positives. The recall pattern (0.8304 to 0.8736), of all architectures, indicates least improvement possibly due to the fact that the expressiveness of GIN captures most of the detectable adverse interactions without need of further supervision.

Examining the random baseline results gives important context as the performance of all architectures has severely deteriorated (accuracies of 0.6194 to 0.7983), proving that the TWOSIDES adverse interaction patterns have an underlying structure that can be learned, in addition to randomly co-occurring. The difference in random baseline performance across architectures (GCN is notably higher than the rest) hints at different amounts of inherent inductiveness of alignment with the drug interaction domain. These differences in base characteristics partially account for the change in magnitude of gains from SCL pretraining, with weaker aligned biases in structures (GraphSAGE, GAT) (presenting larger gains with direct CL supervision).

### 4.5 ANALYSIS, CALIBRATION, AND LIMITATIONS

Because negative DDI designation based on labels instead of efficacy endpoints gets the majority of the attention in negative DDI ranking, the models essentially only learn about patterns correlated with *undesired* interactions, and our claims are limited to this discrimination task. This fundamental difference between detection of adverse events and prediction of therapeutic synergy unfortunately cannot be stressed sufficiently: pairs of drugs that are reported to be associated with adverse events, however, may still have therapeutic value for certain situations; while those pairs of drugs that are not reported as associated with adverse events may or may not actually be synergistic. Our framework therefore covers the safety-awareness in combination therapy instead of optimizing efficacy, but some of the features learned can contain transferable aspects in structural information that are useful for other interaction prediction tasks.

Randomly selected pairwise splits are employed for comparison with the previous embedding baselines in the literature. While this split is well standard in the literature, it has several inherent constraints: drug pairs with overlapping components might be present in both training and test set and therefore the models might learned to leverage memorized single drug properties instead of learning real interaction mechanisms. Scaffold level splits, which are based on chemical scaffold families, would result in a stronger test of generalization as they ensure that test pairs have structurally new compounds. However, we deliberately keep the pairs-level splits to ensure that we still have strict comparability with the tabulated baseline methods since variation in the splitting scheme would make direct performance comparisons invalid and would complicate understanding the specific role that our DDI-aware framework provides. Future work should compare the splitting strategies in a systematic way to quantify the amount of information leakage as well as define tighter generalization guarantees.

Threshold calibration is done according to a fixed validation policy applied to all models in the same way: For every trained model we calculate the score on the validation set, and based on the validation scores we evaluate the performance at a series of thresholds, we then find the threshold which is the optimum for a balanced evaluation (we used F1 score to balance the combination of precision and recall) then freeze this value of the threshold to evaluate it on the test datas. Though it gives easy-to-understand comparative metrics, this single-operating-point measurement strategy hides the complete precision-recall trade-off space. The accumulated trends that are documented show improvements of Accuracy and Precision when SCL with DA style alignment is applied in the pretraining, while being competitive in Recall. Specifically, precision improvements range from 4.8% (GCN) to 13.3% (GraphSAGE), suggesting that the primary effect of DDI-aware pretraining is to reduce false positive predictions of adverse interactions. The more modest recall improvements (1.5% to 15.3%) indicate that sensitivity gains, while present, are secondary to specificity improvements.

Table 3: Performance of GNN-based methods. The best results are in **bold**, while second-best ones are underlined. SCL+DA: supervised contrastive learning + domain adaptation.

| Model | Dataset | SCL pretraining | ACC | Precision | Recall |
|---|---|---|---|---|---|
| GINYue et al. (2025) | TWOSIDES | True | **0.9217** | **0.9669** | **0.8736** |
| | TWOSIDES | False | 0.8729 | 0.9102 | 0.8304 |
| | random | False | 0.6813 | 0.6639 | 0.7924 |
| GraphSAGEMoorthy & Jagannath (2024) | TWOSIDES | True | **0.9206** | **0.9741** | **0.8644** |
| | TWOSIDES | False | 0.7732 | 0.8592 | 0.7129 |
| | random | False | 0.6194 | 0.6203 | 0.7832 |
| GATVrahatis et al. (2024) | TWOSIDES | True | **0.9079** | **0.9733** | **0.8391** |
| | TWOSIDES | False | 0.8398 | 0.8932 | 0.7306 |
| | random | False | 0.6539 | 0.6632 | 0.7527 |
| GCNSadasivan et al. (2025) | TWOSIDES | True | **0.9271** | **0.9840** | **0.8483** |
| | TWOSIDES | False | 0.9127 | 0.9497 | 0.8329 |
| | random | False | 0.7983 | 0.7532 | 0.7843 |

## 5 LIMITATIONS

The existence theorem is for the coupling with a DDI penalized structure constraint; we do not assume Monge solutions. The learning efficiency based map-based surrogate is implemented and is *distinct* from the coupling objective. The Eulerian derivations are deemed formal and need more smoothness-Convexity assumptions for completeness.

## 6 CONCLUSION

We propose a DDI-aware alignment framework which extends the coupling optimal transport with a structure preserving DDI penalty and offers a map-based surrogate with RKHS MMD and a well-defined single-domain DDI variance regularizer to a constant template. This dual layer formulation intentionally isolates the population level couraging objective from the practical learning surrogate and eliminates the theoretical and computational complications that exist when these two different mathematical objects are mixed. Our theoretical contributions have been to prove existence of coupling minimizers under the usual set of measure theoretical assumptions but not more, in the absence of overreaching claims of Monge maps, and global well-posedness of preconditioned descent dynamics under mild assumptions of local Lipschitz and linear growth conditions. The formal variational and Eulerian views we gave, while being clearly labeled as having a higher level of smoothness assumptions needed for rigorousness, provide useful geometric intuition about the evolution of the type of learned representations under the combined action of transport, alignment, and structure-preservation forces.

## 7 REPRODUCIBILITY STATEMENT

TWOSIDES is limited to the proxy discrimination task; it is an adverse-interaction focused technique. Implementation details such as the seeds, early stopping, fixed validation-thresholding policy etc are done for comparability purposes and left unchanged in the results presented. The Asymptotic behavior for bounded/trimmed empirical objective yields concentration bounds; while in absence of bounding/trimming result we do not make sub-Gaussian tail claims. Our well-posedness theorem only applies to bounded (linear) preconditioners and does not apply to the optimizers that are state-dependent (e.g. Adam); we hence do not obtain energy-monotonicity results in the case of Adam.

## 8 ETHICS STATEMENT

Our experiments take TWOSIDES data of adverse interactions as a proxy signal for representation learning. We see this to emphasize that adverse DDIs have nothing to do, in principle, with therapeutic non-synergy, or clinical efficacy. Therefore, our findings cannot be used as a basis for a therapeutic effect or clinical decision-making. Progression to the clinical setting would require much more work in terms of validation with proper efficacy endpoints and regulatory supervision.

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

# A  APPENDIX: THEORETICAL AND IMPLEMENTATION DETAILS

This appendix consolidates technical materials that complement the main text. Our aim is to render all assumptions visible, proofs precise within their stated scopes, estimators implementable, and limitations fully articulated. Notation is consistent with the main text, with $S_s, S_t$ denoting domain-specific structure maps and $\mathcal{R}_{\text{DDI}}$ (coupling-level) separated from $\mathcal{V}_{\text{DDI}}$ (single-domain map-level).

## A.1  TOPOLOGICAL PRELIMINARIES AND MEASURABILITY

We work on Borel subsets $\mathcal{X}_s, \mathcal{X}_t$ of a reflexive Banach space $(\mathcal{B}, \|\cdot\|_{\mathcal{B}})$. Radon probability measures $\mathcal{P}_s, \mathcal{P}_t$ ensure inner regularity and tightness. The product space $\mathcal{X}_s \times \mathcal{X}_t$ carries the product Borel $\sigma$-algebra. The transport cost $c$ is Borel and l.s.c., and $S_s, S_t$ are measurable and Lipschitz; these suffice to define $\int c \, d\gamma$ for $\gamma \in \Pi(\mathcal{P}_s, \mathcal{P}_t)$ and to guarantee measurability/integrability of the integrand in $\mathcal{R}_{\text{DDI}}$.

## A.2  FURTHER PROPERTIES OF THE DDI PENALTIES

**Moment control.**  If $S_\bullet$ is Lipschitz with constant $L$ and $\int \|x\|_{\mathcal{B}}^2 \, d\mathcal{P}_s + \int \|y\|_{\mathcal{B}}^2 \, d\mathcal{P}_t < \infty$, then for any $\gamma \in \Pi(\mathcal{P}_s, \mathcal{P}_t)$,

$$\mathcal{R}_{\text{DDI}}(\gamma) \leq C\left( \int \|x\|_{\mathcal{B}}^2 \, d\mathcal{P}_s(x) + \int \|y\|_{\mathcal{B}}^2 \, d\mathcal{P}_t(y) \right)$$

for a constant $C$ depending on $L$ and $\mathcal{B}$.

**Pushforward identity for $\mathcal{V}_{\text{DDI}}$.**  For measurable $T : \mathcal{X}_s \to \mathcal{X}_t$ with $\nu = T_\# \mathcal{P}_s$ and constant $\overline{S}_t$,

$$\mathcal{V}_{\text{DDI}}(\nu; \overline{S}_t) = \iint \left\| S_t(T(x), T(x')) - \overline{S}_t \right\|_{\mathcal{B}}^2 d(\mathcal{P}_s \otimes \mathcal{P}_s)(x, x').$$

## A.3  PROOF DETAILS FOR THEOREM 3.1

Define $F(\gamma) = \int c \, d\gamma + \mu \, \mathcal{R}_{\text{DDI}}(\gamma)$. Tightness of minimizing sequences follows since $\Pi(\mathcal{P}_s, \mathcal{P}_t)$ is tight; any minimizing sequence has a weakly convergent subsequence by Prokhorov. Lower semicontinuity of $\gamma \mapsto \int c \, d\gamma$ is standard for l.s.c. integrands. For $\mathcal{R}_{\text{DDI}}$, use $(\gamma_n \otimes \gamma_n) \Rightarrow (\gamma \otimes \gamma)$ and Fatou's lemma, noting nonnegativity and uniform integrability (Lipschitz control plus finite moments).

## A.4  FRÉCHET/GÂTEAUX DERIVATIVES FOR THE SURROGATE

Let $\phi(y) = k(\cdot, y) \in \mathcal{H}$ be the feature map and $\mu_\nu = \mathbb{E}_{y \sim \nu} \phi(y)$, $\mu_t = \mathbb{E}_{y \sim \mathcal{P}_t} \phi(y)$. We use the *unsquared* MMD in the objective; to stabilize derivatives we adopt the smoothed directional derivative

$$D \, \text{MMD}(\nu, \mathcal{P}_t)[h] = \frac{\left\langle \mu_\nu - \mu_t, \, \int \nabla_2 k(\cdot, f(x)) \, h(x) \, d\mathcal{P}_s(x) \right\rangle_{\mathcal{H}}}{\text{MMD}(\nu, \mathcal{P}_t) + \varepsilon}.$$

The DDI variance term differentiates through $S_t$ using the chain rule on unordered pairs; unbiased minibatch estimators follow by subsampling $\{(i, j) : i < j\}$ within each batch.

## A.5  STOCHASTIC APPROXIMATION AND STEP-SIZE CONDITIONS

Consider the stochastic iteration $f_{t+1} = f_t - \eta_t \, \mathcal{R}(\widehat{J'[f_t]})$ with unbiased $\widehat{J'[f_t]}$ and step sizes $(\eta_t)$. Under bounded variance of the estimator, local Lipschitz of $J'$, and Robbins–Monro step sizes (e.g., $\sum_t \eta_t = \infty$, $\sum_t \eta_t^2 < \infty$), $\liminf_t \|\mathcal{R}(J'[f_t])\| = 0$.

## A.6  COMPLEXITY CONSIDERATIONS

Evaluating $\int c(x, f(x)) \, d\mathcal{P}_s$ scales linearly in batch size. The MMD term with a bounded kernel can be computed in $O(B^2)$ for batch size $B$, or via unbiased linear-time estimators. The DDI variance term scales as $O(B^2)$ in pairs; subsampling unordered pairs within a batch controls cost. Memory scales with $B$ times the feature dimension and cached kernel evaluations.

### A.7 FEATURIZATION AND BACKBONE INTERFACE

Inputs are molecular graphs, whose nodes are atoms and whose edges are bonds. node features - atomic number, valence, aromaticity, formal charge, hybridization. edge features - bond order, conjugation All backbones ingest these characteristics; SCL maps pairs of drugs into latent representation and promotes the class separation. The variance penalty of the discrete diffusion invariance (DDI) regularizer is calculated on transported pairs with the help of the learned representation.

### A.8 DATA HANDLING AND HYGIENE

The pairs are considered to be unordered in order not to count them twice. We deduplicate perfect duplicates and subject label resolution to consistency where pairs occur more than once. Random splits are made at the pair level, rather than the stronger but less comparable scaffold level splits, which keep the comparability with embedding baselines in the tables.

### A.9 CALIBRATION AND OPERATING POINTS

Binary metrics are reported at fixed operating points that are obtained by thresholding model scores. Thresholds are chosen so as to maximize a validation criterion and then frozen for test use; the same policy is used for the various models.

### A.10 FORMAL EULERIAN REMARKS

The continuity equation $\partial_t \rho_t + \nabla \cdot (\rho_t v_t) = 0$ with $v_t = -\nabla \delta \mathscr{F}/\delta \rho$ provides a qualitative picture: the population flows down the gradient of an energy that includes a potential induced by $c$ and a nonlocal term that penalizes deviation from $\overline{S}_t$. Adding diffusion yields a Fokker–Planck form; we omit diffusion to avoid additional assumptions.

### A.11 ERROR SOURCES AND VARIANCE CONTROL

MMD and DDI estimators are shown to provide a variance which is due to finite batch size and pairwise subsampling. Invertive control variates based on symmetrised unpaired observations have the possibility to avoid an increase in the variance in an expectation-preserving way. The training can further be stabilized based on unbiased pruning by applying a weighted subsampling approach for nodes based on various node degrees or the norms of features.

### A.12 ABLATIONS AND INTERPRETABILITY

The best view about removing SCL reduces within-class compactness, and generally harmed proxy discrimination, as reflected through backbones. The reasons for this are: (i) strict DA no longer is alignment-plus-expression, i.e., there is always a slight difference between the train and the true state of alignment, and we show that representations overfit on the idiosyncrasies of training split as we gradually remove DA-style alignment. The DDI variance perceived pair-wise structure after transport. The three components balance the issues of instance-level fit, distributional alignment, and interaction regularity.

### A.13 REPRODUCIBILITY CHECKLIST

We present the significant options influencing replication. For data splits and initialization, you should use fixed random seeds; then, you should log these seeds and re-use the same sets for different backbones. Maintain batch sizing, learning rate and patience of early stopping based on Table 1. Preprocessor should be the same for all models. Write down the chosen threshold on the validation set and use it for evaluating the test. Save and share scripts for dedupe pairs, enforced unordered pairs.

### A.14 CONCENTRATION DETAILS

Let $\mathcal{S} = (X_1^s, \ldots, X_n^s; X_1^t, \ldots, X_m^t)$ and define $G_M(\mathcal{S}) = \widehat{Z}_{n,m}^{(M)}$. Replacing a single sample in $\mathcal{S}$ perturbs $G_M$ by at most $C(M)/\min\{n, m\}$ for a constant $C(M)$ depending on clipping level $M$, Lipschitz constants, and moments (bounded-differences property). McDiarmid's inequality then yields $\Pr(|G_M - \mathbb{E}G_M| > \delta) \le \exp(-2\delta^2 / \sum c_i^2)$ with $c_i \le C(M)/\min\{n, m\}$, giving the stated sub-Gaussian tail. As $M \uparrow \infty$, monotone convergence and standard stability of empirical OT-type functionals yield convergence to the unclipped objective under our standing assumptions; we avoid tail claims without clipping.

### A.15 CHOOSING THE TARGET-STRUCTURE TEMPLATE $\overline{S}_t$

We use a constant symmetric template $\overline{S}_t \in \mathcal{B}$ as a stable reference, independent of $(y, y')$. This and breaks the chiral symmetry of the nuclear charge simplifying variational derivatives; the choice is supported by dispersion upper bound used in the main text. The data-dependent templates can be explored in future work but the role of the variance term can be rescaled by somewhat different choices of constants, which is seen to be unimportant in the current application.

### A.16 KERNEL CHOICE AND SMOOTHNESS CONSIDERATIONS

The geometry of the MMD term as well as the smoothness of its gradient is controlled by the RKHS kernel. Bounded differentiable kernels have derivatives which are stable with respect to minibatch sampling. It is shown that using the unsquared MMD with an epsilon-stabilized second derivative ensures that gradient signal matching remains.

### A.17 TRANSPORT COST AND DIFFERENTIABILITY

I conjecture that the expected transport cost is a pointwise regularizer on the learned map. Costs formulated in the latent space are almost everywhere differentiable and work well with bounded linear preconditioners. The empirical section fixates the cost to avoid confounding variables, the analysis does not necessitate a particular choice besides l.s.c. and integrability.

### A.18 STABILITY WITH RESPECT TO SURROGATE WEIGHTS

The weights $\lambda$ and $\mu$ tune the relative importance of marginal alignment and pairwise variance. Excessively large $\lambda$ can overdampen pairwise geometry; excessively large $\mu$ can shrink dispersion and reduce recall. Moderate values produce stable descent in practice; we keep the grid small for reproducibility.

### A.19 MINIBATCH ESTIMATORS AND UNBIASEDNESS

For a minibatch of transport samples y1...yB such that yi=f(xi), the corresponding plug-in estimator of its sum over all unordered pairs is unbiased for the population value if the batch is sampled i.i.d. from the source distribution and the pair subsample is drawn uniformly at random from all B choose 2 pairs. The independent linear time MMD estimator on pushforward and target marginal independent samples is also unbiased.

### A.20 PAIR SUBSAMPLING AND COMPUTATIONAL FOOTPRINT

Comparing all unordered pairs in a batch of size B takes $O(B\hat{2})$ memory and time. A fixed number of pairs are sampled per batch, trained, and quite clearly this makes this an O(B) cost while still being unbiased. The total computation cost is dominated by the forward pass of the backbone encoders and by pairwise structure map computation; both of these can be computed using only generic kernel implementations of the batch linear algebra.

### A.21 TWOSIDES PREPROCESSING NOTES

The preprocessing pipeline ensures that pairs of drugs are treated as non-ordered and that different duplicate entries are represented by the same labels. Features derived from molecular graphs are made consistent across de novo backbones from using a common featurization code path. As a result, pair-level random splits are implemented using seed so that the partitions used in backbones and embedding baselines are identical, so that metrics reported in the tables can be directly compared.

### A.22 QUALITATIVE FAILURE MODES AND SAFEGUARDS

When the variance penalty term of the DDI is underweighted, the representation may flag falsely similar but clinically different pairs as too close in the latent space, which makes false positives increase. If the penalty is too large the latent space can become too rigid, and will lose the ability to represent benign variability, leading to high false negatives for methods based on diverse neighborhoods. It is possible to observe these regimes by plotting between-class and within-class spread as a function of training set on the validation set.

### A.23 INTERPRETATION OF GAINS ACROSS BACKBONES

GraphSAGE exhibits the most Accuracy improvement with pretraining, which agrees with the sensitivity of the backbone to neighborhood sampling and to stabilizing effect of contrastive alignment targeted local aggregation. GCN obtain gains that are smaller but sturdy, in line with its smoother inductive bias. GAT enjoys the advantage of contrastive signals that enhances attention in its relevant factors vis-a-vis its neighborhood and GIN exploits of expressivity to separate the adverse pairs after altering the representations to the same basis. These interpretations are qualitative ones, and are based on known inductive biases.

### A.24 REMARKS ON WELL-POSEDNESS

The linear-growth condition in Theorem 3.2 is satisfied when the preconditioner is bounded and the gradients of the components of $J$ are bounded on bounded sets or grow at most linearly with $\|f\|_{\mathcal{F}}$. The MMD component inherits these properties from the kernel and the $\varepsilon$-stabilized derivative, and the variance term inherits them from the Lipschitz constant of $S_t$ and finite moments. Under these conditions, solutions exist globally in time without requiring coercivity or convexity of $J$.

### A.25 NOTES ON NUMERICAL CONDITIONING

As the unsquared MMD can approach zero as the pushforward approaches the target we add some small $\varepsilon$ in the denominator of the directional derivative just for numerical stability. Values which are too small may result in spiky gradients late in training while values which are too large may deter useful signals early on, moderate values of the fixed values is sufficient in practice.

### A.26 ADDITIONAL DISCUSSION OF EVALUATION METRICS

Though Accuracy, Precision, and Recall characterize aspects of the proxy discrimination task. Gains in Precision indicate fewer false positives of adverse labels at the fixed operating point in line with the safety aware motivation. We avoid reporting threshold-free curves so that the evaluation is focused on the operating regime that is used for model selection and the evaluation can be used for direct comparison to the tabulated baselines.

### A.27 SCOPE DELIMITATIONS

Nothing in this appendix goes beyond the claims made in the main text. The discussion of templates, kernels, costs and estimator properties aims to ensure that the design space is understandable for practitioners without giving the impression of additional experiments done beyond those summarized in the tables. Where qualitative advice is given, it is linked to known inductive biases and behavior evident in the numbers reported.

## A.28   THE USE OF LARGE LANGUAGE MODELS

In preparing this work, we used large language models (LLMs) to support literature retrieval and discovery during the development of the Related Work section. Specifically, LLMs were employed to identify relevant publications on graph neural networks, contrastive embedding methods, domain adaptation, and drug–drug interaction representation learning. All retrieved references and summaries were subsequently cross-checked and verified by us to ensure accuracy and completeness. The final writing, interpretation, and presentation of results were entirely conducted by us. Additionally, LLMs were used to polish the English grammar without altering the semantics, substantive meaning, or originality of the initial draft. Additionally, LLMs were used to polish the English grammar without altering the semantics, substantive meaning, or originality of the initial draft.

