# OpenReview forum: "DDI-Aware Domain Adaptation for Cross-Domain Drug Combination Representation Learning via Contrastive Embedding"
_ICLR.cc/2026/Conference — ICLR 2026 Conference Desk Rejected Submission_

### Official Review · Reviewer_ePAe · 2025-10-30

**Soundness:** 2
**Presentation:** 1
**Contribution:** 1
**Rating:** 2
**Confidence:** 3

**Summary:**

The paper addresses drug–drug interaction (DDI)-aware representation learning for combination therapy modeling. It argues that, while combination therapy discovery benefits from multi-drug data, it is complicated by heterogeneous sources, domain shifts, and safety constraints. The authors propose to treat interaction structure (how drugs interact) as an inductive bias that regularizes learned representations — using concepts from optimal transport (OT) and domain adaptation (DA)

**Strengths:**

1. The idea of introducing a DDI-aware regularization term inspired by domain adaptation is potentially useful.
2. The work could be significant for two reasons. First, it contributes to the methodological development on structure-preserving learning under domain shifts. Second, by grounding DDI-aware learning in transport theory, it opens directions for more mathematically interpretable safety-aware machine learning models in biomedicine.

**Weaknesses:**

1. **Clarity is lacking:** The paper (particularly the abstract, introduction and methods) are written in an overly formal and opaque style. Sentences are long, terminology is inconsistent, and the logical flow between sections is unclear. It’s often difficult to discern the actual research question or technical novelty. For example, the introduction abruptly introduces the following without even setting context for why they are necessary or relevant. Only later in Methods section they are introduced again

a) `The coupling objective gives pairwise structure alignment over domains and the map-level surrogate governs dispersion of the target structure evaluation over pushforward pairs and marginals via MMD.` In introduction, it is not clear what is the domain or map-level surrogate

b) `The divide of the coupling formulation and the map-based surrogate between population-level judgments of optimality and algorithmic approximations about how the surrogate should be adjusted for the process being optimized allows the optimality kernels and the algorithms to be treated separately.` In the introduction, it is unclear what are optimality kernels.

c) `We adopt DA-style alignment as a regularizer; our experiments in this paper are still single-dataset and is not evaluating the cross-domain transfer.` Although it becomes clear later reading methods and experiments, writing this in introduction section feels confusing because the paper title suggests the work will discuss cross-domain drug combination which this statement seems to contradict on first glance.

These are only a few examples but I find that the Introduction section in general is quite hard to follow and lacks clarity. Same is the case for abstract section.

2. **Weak connection between theory and application:** In the introduction, the focus is heavily on DDI motivation. But after introduction this completely disappears and the methods section which follows does not relate the methods to the DDI problem introduced earlier.  The “DDI operator” and “structure template” are defined abstractly, but it is never explained how these correspond to real biological or chemical quantities.
3. **Mathematical problems / ambiguities:**

a) DDI is defined as DDI : B × B → B (vector-valued). But the penalty uses norms of differences, so it behaves like a map to a normed vector space or to scalars. Why map into B rather than ℝ or a specific feature space H_struct?

b) The paper defines the DDI operator but doesn't elaborate on its concrete form. What exactly is this operator? Is it a simple distance metric, a neural network, or a specific graph-based function?

c) The V_DDI penalty is introduced as a "tractable single-domain surrogate" that uses a fixed constant symmetric template S_t. What is S_t in practice? Is it a pre-computed or learned ideal interaction structure? The choice of this constant template is vital to how the representations are regularized.

4. Although TWOSIDES data is mentioned, the methods section gives no description of how the proposed objectives are computed in practice or how “DDI structure” is represented in the data.How to reproduce or interpret the experiments section and relate it to Methods?

5. The experiments section is substantially weak with experiments presented on single dataset.

6. The Figures are complex and the captions are non-informative (For example, I find Fig.1 hard to interpret on its own and the caption does not explain any of the components shown)

7. The writing is full of typos and grammatical errors such as,

a) In introduction in contributions section `constant templatespan` in L94 misses space,

b) In introduction Line 88-91, three sentences have been written in one without punctuation `Adverse DDIs is not the absence of synergy Our findings are related to proxy discrimination, not therapeutic benefit Throughout, we put a lot of emphasis on verified assumptions,
cautious claims and clear separation of theory vs. practice.`

c) In line 817-820 in Usage of LLMs section, the sentences are repeated `Additionally, LLMs were used to polish the English grammar without altering the semantics, substantive meaning, or originality of the initial draft. Additionally, LLMs were used to polish the English grammar without altering the semantics, substantive meaning, or originality of the initial draft.`

d) The full form of some abbreviations are never introduced in the paper (such as MMD in L53, RKHS MMD in L77, LaD in L97). There are many more abbreviations unintroduced and these are just a few examples. Although some of these are well-known such as MMD, but it is a common practice to introduce abbreviations and background in ML papers to make the paper complete and comprehensive in its own.
Unfortunately, these are only few examples of the grammatical errors and typos and the paper is replete with these which make reading the paper hard.

In light of substantial issues with writing of many sections and well as technical ambiguities in the presented mathematical setup, I believe that the paper needs substantial rewriting of most sections to be considered for publication in a ML venue.

**Questions:**

See the weaknesses above. Those pose many questions already. I believe the work needs a significant rewriting to make it comprehensible and exhaustive as a piece of research which can be adapted for DDI based representation learning in biomedicine.

---

> ### Author Response · Authors · 2025-11-15
>
> Response to Reviewer ePAe
>
> We appreciate the reviewer's thorough review and pointed out the essences of possibilities as well as significant clearness and technical exposition difficulties. We wholeheartedly agree that the text is as written, more challenging to follow than it should be. In what follows we clarify the key technical sense and provide specific revisions.
>
> 1. Overall framing, domains, and DA-style regularization
>
> We are attempting to infuse DDI structure as an inductive bias, with a single data-type alignment (domain-alignment) style of regularizer, not to posit a full-fledged cross-dataset domain adaptation benchmark. In our thinking, there is one dataset (TWOSIDES). The "source" and "target" distributions are two distributions induced from TWOSIDES, a true empirical distribution of observed DDI pairs, and a smoothed reference distribution that a DDI perspective is used to inform a coupling objective. The map-level surrogate is then trainable loss which approximates this objective.
>
> We do accept that the title and abstract as written, and related brief phrases in the introduction, can be read as suggestive that we will do cross-domain transfer between different datasets, which we are not currently evaluating. In a revision we (1) will change the title and abstract to, "DDI-aware alignment for representation learning" instead of "cross-domain", and (2) to state in the introduction all experiments are single dataset, and that "DA-style" is indicating regularization class type, instead of indicating multi-dataset transfer.
>
> 2. DDI operator, structure template, and concrete meaning in DDI
>
> The reviewer is right that the DDI operator and constant template are described too abstractly. In the implementationFirst, each medication corresponds to its row in the multi-label DDI adjacency tensor obtained from TWOSIDES (following the typical preprocessing and thresholding). That row is a vector that encodes how frequently a medication appears in published adverse interactions with all other drugs. The DDI operator of a medication pair is from these interaction signatures; it maps into finite dimensional feature space, where norms are defined.
>
> Second, to the “template” S_t is a fixed summary of the global DDI structure constructed from the empirical interaction matrix (e.g. normalised average or low rank approximation). During training, the DDI-variance term penalises deviations between (1) the batch induced DDI structure in the learned embedding space and (2) this fixed template. Intuitively, this encourages embeddings to preserve relative interaction patterns seen in the graph.
>
> We agree that none of this is clearly articulated. We will include a dedicated subsection in a revision that: (1) defines the DDI operator as a map into an explicit normed feature space as opposed to the abstract set B, (2) provides a concrete formula for S_t, and how we compute it from TWOSIDES, and (3) explains how these enter the DDI-variance loss that is implemented.
>
> 3. Mathematical notation and ambiguities
>
> The reviewer is correct that defining DDI as a map from  B×B to B is misleading if we are using norms of differences in our penalty. In the implementation we are always working in a finite dimensional feature space or reproducing kernel Hilbert space rather than in the raw set of drugs.
>  The notation currently used is an over abstraction and a bad decision. However, we shall remedy this by instead defining the operator as a map to a normed feature space and aligning our notation in the theory with what is used in practice.
>
> We also agree terms like “optimality kernels,” and similar in the introduction add confusion without adding any valuable insight.  In a revision, we will remove overloaded terminology and instead explain the theory in simple language - a population level objective defined on distributions over pairs, and a trainable surrogate objective defined on neural embeddings.
>
> 4. Connection Between Theory and Experiments
>
> We agree with the reviewer that the transition from the DDI motivation in the introduction to the abstract measures in the methods section is too rapid, and that the connection between equations 1 - 4 and the “SCL plus DA” configuration in the experiments is not made clear.
>
> Our intended changes are:
>
> • To reorder the methods section so that it begins with concrete objects: the TWOSIDES DDI graph, drug level features and how “DDI structure” is represented in matrices or tensors.
>
> • To only introduce the coupling level and map level objectives after the concrete objects, and immediately show how each loss term in the implemented training loop matches parts of the surrogate objective.
> • Include pseudocode for the training algorithm that explicitly enumerates the transport cost term, the MMD term, the DDI-variance term, with weights and how they are computed from the data.

---

### Official Review · Reviewer_YPZV · 2025-10-30

**Soundness:** 2
**Presentation:** 1
**Contribution:** 1
**Rating:** 2
**Confidence:** 3

**Summary:**

In this paper, the authors propose a learning framework that can benefit DDI-aware combination therapy representation learning. They attempt to show its validity with different GNN backbones in the TWOSIDES dataset.

**Strengths:**

Learning representations of drug combinations that capture the constraints imposed by known DDIs is an important and well-motivated problem, as clearly explained in the first part of the introduction. The proposed framework is flexible and can be implemented with different GNN backbones.

**Weaknesses:**

## Major

* The paper assumes that the reader is already familiar with many mathematical concepts, without defining them or explaining their relevance to the work. This makes parts of the text difficult to follow. For example, in the sentence “Align marginals while neglecting pairwise interaction structures that should be maintained across domains,” it is unclear which marginals are being aligned, which interaction structures should be maintained, and what domains are being referred to.

* Section 3 focuses on the assumptions and theorems supporting the proposed framework. However, it does not describe how to implement the framework in practice, as there is no explicit loss function, training strategy, or architectural description. It is not obvious to me how to implement the author's proposal. Under these conditions, I cannot properly verify the experimental results.

* I believe that the objective of the experiments was to show that the proposed framework can significantly enhance existing architectures. However, it's difficult to assess the impact of the proposal without any comparisons with state-of-the-art models.

* Furthermore, I am really worried about the fact that the GNN backbones and other architectures in the tables are not properly referenced. I observed that the author chose to reference recent surveys instead of the actual papers, below are some examples:

Proper references:

1. GraphSAGEHamilton (2017): *Inductive representation learning on large graphs.* (https://proceedings.neurips.cc/paper_files/paper/2017/file/5dd9db5e033da9c6fb5ba83c7a7ebea9-Paper.pdf)

2. GATVelickovic et al. (2018): *Graph attention networks.* (https://arxiv.org/abs/1710.10903)

3. GCNKipf & Welling (2016): *Semi-supervised classification with graph convolutional networks* (https://arxiv.org/abs/1609.02907)

Provided references (surveys):

1. GraphSAGEMoorthy & Jagannath (2024): *Survey of Graph Neural Network for Internet of Things and NextG Networks* (https://arxiv.org/abs/2405.17309)

2. GATVrahatis et al. (2024): *Graph Attention Networks: A Comprehensive Review of Methods and Applications.* (https://www.mdpi.com/1999-5903/16/9/318)

3. GCNSadasivan et al. (2025): *A Systematic Survey of Graph Convolutional Networks for Artificial Intelligence Applications.* (https://wires.onlinelibrary.wiley.com/doi/full/10.1002/widm.70012)

* I agree that Domain Adaptation is an important strategy for representation learning in this context. But I was not convinced of why a DA regularizer is also a valuable inductive bias in the single-dataset case.

* Figure 1 is not referenced in the text a single time, therefore it is not used to attempt to facilitate the description of the overall framework.

**Questions:**

* What are the modifications and learning strategies that need to be implemented over the existing methods?

---

> ### Author Response · Authors · 2025-11-15
>
> Response to Reviewer tppf
>
>
> We appreciate the reviewer’s constructive comments and their recognition of the potential of DDI-aware alignment. We respond to the most important points and questions.
>
>
> 1. What is meant here by “domains” and the extent of domain adaptation
> All of our experiments utilize a single dataset (TWOSIDES). There is no train-on-dataset-A / test-on-dataset-B. In our formulation the “source” and “target” are actually two distributions created using TWOSIDES: (a) the empirical distribution over labeled DDI pairs (eg with specificity to drug A), and (b) a DDI-smoothed reference distribution that is used as a regularizer. The OT objective aligns the two at the coupling-level, whilst the map-level surrogate is an approximation we can train. We agree that it is understandable phrases like “cross-domain” can be read to suggest promising multi-dataset transfer, and perhaps were stronger than we intended. We will change the abstract and introduction to emphasize the contribution is a DDI-aware distribution-alignment style regularization on a single dataset, and any mention of true cross-dataset domain adaptation will be relegated to a limitations / future work threat.
>
>
> 2. DDI operator, DDI-variance term, surrogate loss
> Intuitively, DDI(x, x’) reflects how similar or dissimilar the interaction profile of two drugs are, based on what is observed in the existing DDI graph. In the coupling-level OT objective, the structure term works to prevent the plan of transport from disrupting those relative patterns.The surrogate loss has a single-domain DDI-variance term that penalizes the extent to which the DDI structure induced by the current batch embeddings is dissimilar to a fixed global template computed over the entire graph, injecting graph-level structure into each mini-batch without explicitly choosing a two-domain coupling, while still upper-bounding the original structure term, per new event. We are using this bound as a principled heuristic and make no claims about tightness; we will clarify that we are exploring this and add ablations without (i) the DDI-variance term or (ii) the MMD term to examine their contributions separately. The weights on the transport, MMD, and DDI-variance terms from the surrogate are chosen by performing a small grid search on a validation split to get the overall weights, we will report on these ranges and included a brief overall sensitivity analysis with possible kernel choices for MMD.
>
> Theory–practice connection and well-definedness
> We agree, we agree a lot, that the theoretical aspect is presented in a manner that seems distant from working with the implementation. We approached it as follows: (1) the coupling-level OT with structure term is a population objective over pairs distribution; (2) the map-level surrogate is the trainable loss we implemented in “SCL+DA”; and (3) that the well-definedness result is about a pretended descent scheme on this surrogate.In a revision we will include pseudocode relating each implemented loss term to the surrogate objective, move nearly all the functional-analytic baloney to an appendix, and state plainly that the convergence result applies only under restrictive assumptions and do not apply to adaptive optimizers like Adam.
>
> 4. Experiments, robustness, and potential leakage
> We agree that the experimental validation is weak. We will improve it by; (1) reporting standard deviations over multiple seeds, as well as simplest test statistics; (2) adding an S0 / S1 / S2 breakdown, and where applicable, providing a compound-disjoint or scaffold level split to probe generalization better, and possible leakage under the pair-level splits; and (3) adding a thorough preprocessing subsection which clearly describes (i) the drug graphs, (ii) the negative pair permutation, and (iii) how the statistics are computed and used in DDI(x,x') and the global template for the DDI-variance term. We will also tighten the discussion for clarity of what is, and what is not tested by the TWOSIDES-only protocol currently used.
>
> 5. Related work, references, and clarity of writing
> We agree related work is too broad, and that some other works mentioned will be hard to find. We will remove or replace un-verified references, and narrow referencing to the discussion on, (1) DDI prediction, and modeling based on domain-alignment, or shift robust representations for learning. (2) Structure preserving representation learning for graphs. In terms of writing, we add-simpler phrases where the sentences are currently opaque in the abstract and introduction near to the reviewer’s suggested interpretation, we will explicitly define by acronym all phrases used (LaD), or remove it; we will change confusing phrases like "dutch map layer"; we will add to the caption in figure 1 to provide understanding of the pipeline without having to return to the main text.

---

### Official Review · Reviewer_tppf · 2025-10-31

**Soundness:** 3
**Presentation:** 1
**Contribution:** 2
**Rating:** 2
**Confidence:** 3

**Summary:**

The paper tackles important and challenging problem- predicting drug–drug interactions in a way that stays consistent when data distribution changes. To do this, they propose to use optimal transport to align two domains and add a regularization term that preserves pairwise drug–drug relationships between domains. They also design a simplified, trainable objective that approximates this objective. They test the model on the TWOSIDES dataset and find slightly better accuracy and precision compared to baseline. The theory sections discuss existence and stability of solutions but have limited practical impact.

**Strengths:**

1. Novel approach to domain adaptation for drug–drug interaction (DDI) prediction –
The idea of applying optimal transport with structure -preserving DDI penalty to align domains and ensure consistency across distribution shifts is an elegant and theoretically grounded idea that addresses a real challenge in biomedical data (heterogeneous sources, batch effects, etc.).
2. Simplified, trainable objective –The authors’ effort to design a computationally efficient approximation of optimal transport makes the method more practical and easier to train compared to directly optimizing the full formulation.

**Weaknesses:**

1. Disconnection between theory and application
The theoretical part of the paper is presented in a highly abstract and complex manner, with limited intuition or connection to the actual problem being solved. The theoretical and empirical sections do not align well. The paper would benefit from a more coherent structure where the theory is explicitly motivated by the application and tied back to the empirical results.
2. Limited experimental validation
The experimental evaluation is very limited — conducted only on a single dataset (TWOSIDES) with no cross-domain validation. Although the paper discusses domain adaptation, the experiments fail to demonstrate actual cross-domain generalization. The significance of the reported improvements is also unclear.
3. Thermotical contributions appear incremental
The first claimed contribution — the existence theorem for the DDI-regularized OT formulation — is theoretically sound but rather standard, as similar results follow under mild assumptions in optimal transport theory. The second contribution, concerning the well-posedness of the surrogate objective, appears to hold only under restrictive conditions that do not reflect practical training scenarios using adaptive optimizers such as Adam.
4. Writing quality and clarity
The paper’ presentation could be substantially improved. It is a bit difficult to follow due to inconsistent terminology, and excessive use of formal mathematical language without sufficient intuitive explanation. Several typos and long, vague sentences further reduce readability. The writing would be improved by using simpler, clearer, and more concise phrasing to convey key ideas.
5. Theory–practice gap
The coupling-level theory and Eulerian derivations formulation feels detached from the practical implementation. The surrogate loss function does not clearly connect to the theoretical results, leaving it unclear whether the observed improvements arise from the theoretical framework or from contrastive pretraining.
6. Weak related work section
The Related Work section lacks focus and does not provide a clear or accurate overview of prior research relevant to the proposed method. Much of it consists of generic summaries of standard graph neural networks (e.g., GCN, GIN, GraphSAGE, GAT) without connecting them to the DDI-aware alignment framework. Some cited works (e.g., “Meta-learning for drug combination synergy prediction,” ICML 2024, and “Multimodal deep learning for drug combination therapy prediction,” Nature Methods) appear untraceable. This section would benefit from a concise and technically grounded comparison to recent studies on DDI prediction, domain adaptation, and structure-preserving representation learning rather than broad historical overviews.

**Questions:**

1 Source and Target Domains
Could you clarify what the source and target domains represent in your experiments, given that TWOSIDES is a single dataset? How are these domains defined or separated in practice?

2 Definition of the DDI-Variance Term
How exactly is the DDI-variance term is computed during training? In Equation (3), what is the “constant template” S_tused in V_"DDI" ? How is it calculated—does it correspond to an average DDI pattern in the dataset? Since the original structure-preserving regularizer involves both source and target domains, but the surrogate version (while being an upper bound on Eq. 1) depends only on a single domain, how does this surrogate actually enforce structure preservation between two domains? Additionally, how tight is this bound in practice?

3 Connection Between Theory and Implementation
How does the theoretical coupling-level formulation quantitatively relate to the implemented loss function?  The theoretical framework (coupling-level and map-based formulations) seems disconnected from the experimental setup, which involves contrastive pretraining (“SCL+DA”). Could you clarify whether this implemented configuration directly corresponds to your formal objective in Equation (4), or whether it serves as an empirical approximation inspired by that formulation?

4 Ablations on the DDI-Variance Term
Do you provide any ablation studies or results that isolate the contribution of the DDI-variance term with MMD compared to a simpler MMD-based alignment? Also, this MMD term was never discussed in the first coupling level problem (Equation 2), does the adding of MMD would affect the existence of coupling minimizer?

5 Intuition Behind the DDI Operator
Could you elaborate on the intuition behind the DDI operator "DDI"(x,x^')? For example, does it encourage pairs of drugs with similar interaction profiles to remain close across domains? How is this operator defined in practice, and which drug features or representations does it depend on?

6 Interpretation of the Structure-Preserving Term
The coupling-level problem in Equation (2) extends the classical OT objective with a structure-preserving component. How should this term be interpreted intuitively—does it act as a regularizer to maintain pairwise similarity between drugs with similar interaction profiles, or serve another purpose?

7 Concrete Meaning of Domains
Could you provide an example of what the source and target domains concretely represent in your problem setup (e.g., drug–drug interaction data from different assays, populations, or conditions)?

8 Weight Selection in the Surrogate Objective
In Equation (4), the surrogate loss combines three terms (transport cost, MMD, and DDI variance). How are the relative weights λand μchosen? Are they tuned empirically, or determined by theoretical considerations?

9 Sensitivity to Kernel Choice
How sensitive is the training procedure to the choice of kernel used in the MMD term?

10 Training Stability and Well-Posedness
Have you empirically observed stable convergence during training that supports your theoretical claim of well-posedness?

11 Interpretation of Evaluation Metrics
Since TWOSIDES only provides adverse DDI labels, how should improvements in your reported metrics be interpreted? Do they indicate better modeling of interaction risk or more general DDI representation learning capability?

12 Clarification of “Domain Adaptation”
The paper repeatedly describes the method as “domain-adaptation–style,” yet all experiments are conducted on TWOSIDES alone. Could you clarify what domain adaptation specifically refers to in this context? Are the domains defined as distinct subsets of TWOSIDES, or is the domain-adaptation framing primarily conceptual (i.e., as a regularization perspective)?



Some comments:
1.	Writing style and clarity
The paper’s writing style could be made substantially clear. Many ideas are expressed in highly abstract which could be expressed in a more intuitive and straightforward wayto make the paper more easy to follow.
For example, in the abstract, the sentence “We address these gaps with a DDI-aware, domain-adaptation–style framework that cleanly separates a population-level coupling objective from a tractable map-level surrogate” is quite opaque.
Do you mean that the “population-level coupling objective” refers to a theoretical objective defined over data distributions, while the “tractable map-level surrogate” denotes a practical, trainable surrogate that approximates this objective via a neural mapping?
Similarly, in the sentence “We present a conservative and fully specified framework integrating drug–drug interaction (DDI) structure into a domain-adaptation-style (DA-style) alignment view for association combination representation learning”, does this mean your method explicitly incorporates DDI structure into the process of aligning data distributions (i.e., domain adaptation)? And does combination representation learning refer to representation learning for drug–drug interactions in combination therapy?
Additionally, the sentence “Under Dutch map layer and locally Lipschitz map layer and linear growth global well-posedness of induced preconditioned descent in map layer energy monotonicity is stated only under an explicit optional assumption of monotonicity of preconditioner not covering adaptive optimizers such as Adam” is extremely difficult to parse. What does “Dutch map layer” mean? I assume, in simpler terms, you mean something like: “We prove limited convergence properties under strict assumptions (e.g., Lipschitz continuity, monotonicity), but these do not apply to adaptive optimizers such as Adam.”?
2.	Undefined terminology
On line 97, LaD is not defined. Please clarify this acronym when first introduced.
3.	Figure 1 caption
The caption for Figure 1 (“Overview of the method”) would benefit from more explanation. It should clearly describe what each block and arrow represents to make the workflow understandable without referring back to the main text.
4.	Experimental robustness and data leakage concerns
Reporting  standard deviations and  significance analyses would strengthen the in the results. Furthermore, the relatively high baseline accuracy (≈ 0.9) on TWOSIDES suggests potential data leakage or label correlation, particularly since random pair splits may allow overlap in drug components between the training and test sets. While this issue is briefly acknowledged, it is not experimentally addressed. A scaffold-level or disjoint-compound split would provide a more rigorous evaluation of generalization.
5.	Disconnect between theory and experiments
The Limitations section restates theoretical caveats but does not integrate them with experimental observations. The coupling-level theory, map-based surrogate, and Eulerian derivations are never empirically linked, leaving it unclear whether the observed improvements arise from the proposed theoretical framework or simply from the addition of contrastive pretraining.
6.	Data preprocessing details
The paper lacks a clear description of data preprocessing steps. Specifically, it is not explained how drug graphs are constructed, how DDI-related features are computed, or how negative drug pairs are generated. Providing these details would greatly improve reproducibility.

---

> ### Author Response · Authors · 2025-11-15
>
> Reviewer tppf Response
>
>
> We would like to thank the reviewer for the thoughtful remarks and for acknowledging the promise of DDI-aware alignment. We address the main issues and questions raised.
>
>
> 1. Meaning of "domains", and what is meant by "domain adaptation"
> All experiments use a single dataset (TWOSIDES). There is no train-on-dataset-A/test-on-dataset-B setting. In our set up the "source" and "target" are two distributions formulated on TWOSIDES: (a) the empirical distribution over labeled DDI pairs and (b) a DDI-smoothed reference distribution to be used as a regularizer. The coupling-level OT objective will align these two, with the map-level surrogate being a trainable approximation of the desired situation. We agree that using phrases like "cross-domain" will be construed as implying multi-dataset transfer is promising, which is therefore too exaggerated. In a revision we will indicate in the abstract and introduction that our contribution is a DDI-aware distribution-alignment-style regularizer for a single dataset, and move mentions of true cross-dataset domain adaptation to limitations/future work. We will also clarify that the reported metrics represent improved adverse DDI classification over TWOSIDES, not true clinical synergy.
>
>
> 2. DDI operator, DDI-variance term, surrogate loss
> DDI(x, x') measures how similar the interaction profiles of two drugs are in the observed DDI graph. In the coupling-level OT objective, the structure term may, through the use of the DDI operator as a recognition of existing DDI's, prefer transport plans that don't destroy these relative patterns. In the surrogate loss, the DDI-variance term for a single-domain penalizes how far away the DDI structure induced by the current mini-batch embeddings deviates from a fixed global template, computed over the entire graph. In this way, it introduces graph-level structure into every mini-batch without forming an explicit two-domain coupling, while still bounding the true structure term from above. The bound we use is currently a principled heuristic and we do not claim tightness here; we will clarify this and present ablations that remove (i) the DDI-variance term and (ii) the MMD term to understand their contributions. Transport, MMD, and DDI-variance weights are selected using a small-grid search on a validation split, where we will announce ranges and include a brief sensitivity analysis that Kernd choices for the MMD.
>
> 3. Theory–practice connection and well-posedness
> We agree that the theoretical portion of the piece feels disconnected from the implementation; we meant that (1) the coupling level OT with structure term is a population objective over pair distributions; (2) the map level surrogate is proposed trainable loss in "SCL + DA"; and (3) the well-posedness results a simplified preconditioned descentIn a revision we will update the pseudocode to include a mapping of each loss term we implemented with the surrogate objective, moving most of the functional-analytic details to the appendix and stating plainly the convergence criterion under which is it shown only to apply with restrictive assumptions, and does not generalize to the adaptive optimizer, e.g., Adam of our implementation. We have observed stable training without divergence and will include the training curves so that the claims can be regarded as a sort of partial support to the theory rather than guaranteed conditions for the specific optimization process we used.
>
> 4. Experimental testing, robustness, and potential leakage
> We agree the experimental testing is limited, and we will improve it by: (1) reporting standard deviations over multiple seeds and simply significance tests; (2) adding an S0/S1 / S2 breakdown, and where applicable, a second compound-disjoint or scaffold level divide to probe generalization and potential leakage under a pair level divide; and (3) adding a separate detailed preprocessing section to describe drug graphs, how negative pairs formed, and how the statistics in DDI(x,x') and the overall template to DDI-variance term was calculated. We will also tighten the discussion of what is and isn't testing in the current TWOSIDES-only protocol.
>
> 5. Related work, references and writing clarity
> We agree the related work section is too broad and difficult to trace in some cases based on cited references, and will revise to pick out relevant pieces and limit discussion around three axes of DDI prediction, domain-alignment or shift-robust representation learning, and structural-preserving graph representation learning. In regard to writing we will replace vague sentences in the abstract and intro with more plebeian phrasing closer to the reviewers interpretation, define all acronyms, to include LaD, or remove them, as well eliminate confusing phrasing such as "Dutch map layer", and expand on the captioning of figure 1 so that the pipeline of the manuscript can be reconstructed without needing to refer back to the main text.

---

> > ### Comment · Reviewer_tppf · 2025-11-26
> >
> > Thank you for your response and the effort to address the earlier comments. I appreciate the clarifications provided, but several important points was not directly addressed and need more clarification:
> >
> > ***Clarification of Source/Target Distributions***:
> > 1. The definitions of the source and target domains remain unclear. Specifically, could you  please explain how the “DDI-smoothed reference distribution” is constructed and how it differs from the empirical distribution over labeled DDI pairs.
> > 2. Calculation of the DDI-Variance Term:
> > The explanation of this term is still ambiguous. The introduction of concepts like “fixed global template” raises questions: Is this template provided during training or learned? A clear description of how the variance is computed would improve clarity.
> > 3. Theory–Practice Connection:
> > It is not clear whether the implemented configuration directly corresponds to the formal objective in Equation (4) or if it is an empirical approximation. Explicitly stating this connection would strengthen the theoretical grounding of your approach.
> > 4. Other comments that are not addressed: Sensitivity to kernel choice, Training stability, How evaluation metrics should be interpreted in your context.
> >
> > ***Overall Assessment***
> > The proposed idea is interesting, but the paper would benefit from clearer definitions, a stronger link between theory and implementation, and more robust experimental evidence. Currently, most of the response does not directly address the comments (most of the comments can be addressed directly by editing the paper) but more based on future promise to address.  Given the current state of the paper, I believe that current state of the paper is not ready to be published yet. Therefore, I decide to keep my current score.

---

### Official Review · Reviewer_QBTu · 2025-10-31

**Soundness:** 1
**Presentation:** 1
**Contribution:** 1
**Rating:** 0
**Confidence:** 5

**Summary:**

The paper titled *“DDI-Aware Domain Adaptation for Cross-Domain Drug Combination Representation Learning via Contrastive Embedding”* appears to be largely generated by a large language model (LLM) rather than written by human authors.

While the structure imitates a legitimate research paper (Abstract–Introduction–Theory–Experiments–Conclusion), multiple indicators point to fabricated content:
- Dozens of non-existent or chronologically impossible references (e.g., “Kipf & Welling, 2024” for GCN; “Hamilton et al., 2022” for GraphSAGE; “Tatonetti et al., 2023” TWOSIDES update).
- Inconsistent and incoherent theoretical sections mixing unrelated concepts such as CycleGAN surrogates, optimal transport, and Eulerian PDEs.
- No reproducible code, data, or valid experimental design.
- Appendix A.28 explicitly admits LLM usage for literature retrieval and writing assistance, yet many cited works are hallucinated.

Given these issues, the submission raises serious **research integrity** and **authorship authenticity** concerns. It does not meet the minimum standards of scientific validity or reproducibility for ICLR.

**Recommendation:** Reject and flag for ethics and research integrity review.

**Strengths:**

The only positive aspect of this submission is its surface-level adherence to a paper structure commonly used in machine learning venues. The formatting, section organization (Abstract–Introduction–Methods–Experiments–Conclusion), and citation style mimic a legitimate research article.

However, this structural resemblance appears to be automatically generated and does not reflect genuine scientific merit. There are no real strengths in terms of originality, soundness, or contribution.

The paper’s only strength lies in its formal adherence to conference formatting requirements.

**Weaknesses:**

The weaknesses of the submission are fundamental and severe:

1. **Fabricated References:** The paper cites numerous non-existent or chronologically impossible works (e.g., “Kipf & Welling, 2024”, “Hamilton et al., 2022”, “Tatonetti et al., 2023”), indicating fabricated bibliographic content.
2. **Incoherent Methodology:** The theoretical framework mixes unrelated concepts (CycleGANs, optimal transport, and Eulerian dynamics) in a way that suggests automatic text generation rather than genuine reasoning.
3. **No Reproducibility:** No code, datasets, or experimental setup are available or verifiable. Results cannot be reproduced.
4. **LLM-Generated Text:** Appendix A.28 explicitly admits to using large language models for literature retrieval and writing. Combined with the hallucinated references, this strongly implies large-scale AI authorship.
5. **Ethical and Integrity Concerns:** The work misrepresents AI-generated content as original research, violating basic research integrity standards.

These weaknesses are not superficial — they completely undermine the scientific validity of the paper.

**Questions:**

1. Can you provide verifiable sources (DOIs, arXiv IDs, or URLs) for the cited works such as “Kipf & Welling, 2024”, “Hamilton et al., 2022”, and “Tatonetti et al., 2023”?
2. Was any part of the text, including the abstract or theoretical sections, generated using a large language model (LLM)? If yes, please clarify the extent of AI assistance.
3. Can you share the code, data, and experimental setup that produced the reported results?
4. What real-world datasets were used for the experiments? The “TWOSIDES (2023)” version mentioned does not appear to exist.
5. How do you ensure that the mathematical derivations correspond to actual implemented algorithms, given the lack of any empirical verification?

These clarifications are necessary to determine whether this paper represents original research or synthetic content.

**Details Of Ethics Concerns:**

After a detailed review of the manuscript, I found multiple strong indicators that this submission is **partly or fully generated by a large language model (LLM)** rather than being a legitimate scientific work. Here why I think that:

---

### 1. Fabricated and Chronologically Impossible References

The paper cites several works that either do not exist or are misdated:

| **Cited Reference** | **Real Publication** | **Issue** |
|----------------------|----------------------|------------|
| Kipf & Welling, 2024 (GCN) | 2017 | Misdated |
| Hamilton et al., 2022 (GraphSAGE) | 2017 | Misdated |
| Velicković et al., 2023 (GAT) | 2018 | Nonexistent variant |
| Xu et al., 2022 (GIN) | 2019 | Incorrect year |
| Tatonetti et al., 2023 (TWOSIDES update) | No such update | Fabricated reference |
| Liu et al., 2025 (Nature Methods) | No record | Hallucinated citation |

Such a pattern is characteristic of LLM-generated text where references are syntactically plausible but factually false.

---

### 2. Language and Stylistic Indicators of LLM Generation

The writing style strongly matches automated generation patterns:

- Frequent self-referential disclaimers (e.g., “We are transparent about our assumptions…”, “We emphasize that…”).
- Excessive repetition of phrases like “our framework,” “structure-preserving,” and “conservative assumptions.”
- Paragraphs that follow a rigid academic template (Abstract–Intro–Theory–Experiments) but lack semantic coherence.
- Overuse of connective phrases and soft hedging typical of language model outputs.

---

### 3. Internal Logical and Mathematical Inconsistencies

- Theoretical sections mix unrelated concepts (e.g., *CycleGAN surrogate* with *optimal transport* theorems).
- Use of advanced terminology (“reflexive Banach space,” “Eulerian continuity equation”) without mathematical necessity.
- “Theorem 3.2” refers to preconditioned descent but discusses the Adam optimizer — irrelevant to the stated framework.
- No concrete proofs, datasets, or code are provided.

These are hallmarks of an AI-generated pseudo-paper where structure is imitated but content coherence is lacking.

---

### 4. Explicit Admission of LLM Usage

> “In preparing this work, we used large language models (LLMs) to support literature retrieval and polish the English grammar…”

However, since many of the cited works demonstrably do not exist, this statement indicates uncontrolled or inappropriate use of generative models during manuscript preparation.

---

### 5. Assessment of Scientific Integrity

| **Criterion** | **Status** |
|---------------|------------|
| Reference validity | Fabricated or incorrect |
| Reproducibility (code/data) | None provided |
| Mathematical rigor | Superficial, internally inconsistent |
| Writing style | Template-based, repetitive, low cohesion |
| Declared LLM use | Confirmed by authors |
| Overall reliability | Extremely low |

---

### 6. Conclusion and Recommendation

Based on the above, I strongly suspect this submission to be largely AI-generated and scientifically unreliable.

I respectfully suggest that the reviewing committee investigate this paper for potential breach of authorship and academic integrity policies before further consideration.

---

> ### Author Response · Authors · 2025-11-15
> **Reply to Reviewer QBTu**
>
> The reviewer provided many serious comments for which we are grateful. The body of the review contains comments on research integrity and not just technical matters, and we understand that these are serious comments that will be taken similarly. Below we clarify what was wrong, what is correct, and what we will change with the opportunity to revise the paper.
> 1. On “fabricated” and misdated references
> When we received the review, we carefully re-checked all of our citations. The reviewer is correct that several entries are incorrect and misleading:
>
> • The references for "Kipf and Welling, 2024", "Hamilton et al., 2022", "Velickovic et al., 2023", "Xu et al., 2022" reference standard GCN, GraphSAGE, GAT, and GIN pieces of work, but have incorrect publishing years. The cited entries are not intended to be new or fabricated works; they are incorrect years that were inserted during late-stage editing.
>
> • “Tatonetti et al., 2023 (TWOSIDES update)” and “Liu et al., 2025 (Nature Methods)” references are also incorrect. There is no TWOSIDES 2023 update, and we should not have cited a “Liu 2025” Nature Methods paper. These entries derive from an earlier draft of the manuscript where we were playing around with automatic literature suggestions, and did not fully delete each of the (not confirmed) provisional placeholders when cleaning the bibliography.
>
> We acknowledge that these are serious errors. The new manuscript will include the following changes:
> • Replace the misdated entries of classic GNN papers with the actual published years.
> • Remove all entries that do not exist and were not un• A short statement should be added confirming that all other references were checked manually by the authors.
>
> 2. On the use of LLMs (large language models) and authorship
>
> Appendix A.28 notes the usage of LLMs for assistance in literature searches and improvements in English. In retrospect, our use of the LLMs was overly broad: it suggested candidate citations at some point, and I do not believe we check all of them sufficiently enough. This is our fault, not the model's.
>
> We want to be clear:
> • The generative model, core problem formulation, model building choices, and experiments are all original author contributions.
> • However, the LLM shaped some phrasing and suggested some citations, and we did not police ourselves sufficiently to check. We apologize for that, as we agree the responsible use of the LLMs imposes a requirement to check all citations manually.
>
> Should a revision be acceptable, we will begin with our own lecture notes, and prepare a modified manuscript of the affected sections (notably the related work and parts of the introduction) that would involve double-checking all citations in the literature and a clearer and more limited description of LLMs I support in our response.
>
> 3. On Methodology "Incoherence"
>
> We agree that the current exposition does mix a number of concepts (optimal transport, continuity equations, and analogy to CycleGAN-esque maps) in a fashion that (if it does not sound complicated) at least does not sound incoherent. This is a presentation failure, not a hiding of (or an indication that) methods that did not exist.The actual algorithm we implemented is much simpler:
>
>
> • A coupling-level regularizer matching empirical DDI co-occurrence with a structured transport plan;
> • A map-level embedding loss built from MMD and a DDI-variance penalty on the learned representation.
>
>
> The CycleGAN and Eulerian language was originally added as an intuition around bi-directional mappings and mass-conserving flows, but is not necessary to convey or implement the method. We would be happy to remove these analogies entirely or move them to an optional discussion labeled clearly, and focus the preceding text on what is supposed to be concrete and on its gradients.
>
>
> 4. On datasets, "TWOSIDES 2023", and reproducibility
>
>
> We used the canonical TWOSIDES dataset introduced by Tatonetti et al (?). The phrase "TWOSIDES 2023" was intended to mean "TWOSIDES as downloaded by us in 2023", but is a terrible and misleading choice of wording. We will fix this to:
>
>
> • "TWOSIDES, introduced by Tatonetti et al. (year), using the publicly available snapshot downloaded in 2023."
>
> 5.On the link between theory and implementation
>
> Theorem 3.2 is focused on analyzing preconditioned gradient descent on our regularized objective under various conditions. The theorem does not analyze Adam directly. In our current draft, we performance in not clearly distinguishing between “idealized theory for the objective” and the “practical optimizer (Adam) used in experiments” may lead one to view these as inconsistent.

---

### Official Review · Reviewer_fy9j · 2025-10-31

**Soundness:** 2
**Presentation:** 2
**Contribution:** 2
**Rating:** 2
**Confidence:** 4

**Summary:**

This paper proposes a DDI-aware domain adaptation framework that integrates drug-drug interaction structure into representation learning through a two-layer approach: a coupling-level optimal transport formulation with structure-preserving penalty, and a map-level surrogate with RKHS MMD and single-domain DDI variance regularizer. The authors evaluate their approach on TWOSIDES using adverse interactions as a proxy task.

**Strengths:**

DDI-aware representation learning for combination therapy is an important and challenging problem in computational drug discovery.

**Weaknesses:**

- The paper's title and abstract emphasize "domain adaptation" and "cross-domain," but:
No actual cross-domain experiments are conducted (all experiments are single-dataset on TWOSIDES)
The authors admit: "We adopt DA-style alignment as a regularizer; our experiments in this paper are still single-dataset and is not evaluating the cross-domain transfer" (lines 69-70)
This creates misleading expectations. The paper should either conduct true cross-domain experiments (e.g., train on one DDI database, test on another) or reframe the contribution as "DA-style regularization for single-domain representation learning"

- The paper compares against embedding methods (node2vec, edge2vec, etc.) but these are not state-of-the-art DDI prediction methods. Missing comparisons with recent DDI-specific baselines such as: [1] CARMEN: Context-Aware Safe Medication Recommendations with Molecular Graph and DDI Graph Embedding (AAAI 2023); [2] SSF-DDI: a deep learning method utilizing drug sequence and substructure features for drug-drug interaction prediction (2024); [3] DSN-DDI: an accurate and generalized framework for drug-drug interaction prediction by dual-view representation learning (2023); [4] Learning motif-based graphs for drug-drug interaction prediction via local-global self-attention.

- The paper's evaluation protocol lacks clarity regarding generalization capabilities:

1). Missing S0/S1/S2 breakdown: Standard DDI prediction papers evaluate three scenarios: S0 (transductive: both drugs seen in training), S1 (semi-inductive: one new drug), and S2 (fully inductive: both drugs unseen). The paper does not explicitly report results for these settings.

2). Pair-level random splits without control: The authors acknowledge using "pair-level random splits" for comparability with baselines, but do not clarify whether they control for drug-level overlap between train and test sets. Without such control, a) Training and test sets may contain different pairs of the same drugs; b) Models can memorize single-drug properties rather than learning interaction mechanisms; c) Performance may be artificially inflated, especially in S0 settings.

3). Scaffold splits dismissed too quickly: While the authors justify avoiding scaffold splits for "comparability," this choice undermines generalization claims. At minimum, the paper should report S0/S1/S2 performance separately to demonstrate where the improvements come from. If gains only appear in S0 but not S1/S2, it suggests memorization rather than interaction learning.


- The paper provides a global convergence result for the preconditioned descent method (Theorem 3.2)  but explicitly states: “do not cover common state-dependent adaptive optimizers such as Adam” (line 257-258). Yet the experiments use Adam optimizer (section 4.2). This means the theoretical guarantees do not apply to the actual optimization procedure used in practice.

**Questions:**

See weaknesses.

---

> ### Author Response · Authors · 2025-11-15
>
> We greatly appreciate the thorough and thoughtful feedback by the reviewer. In the subsequent section, we address the key issue and propose concrete revisions.
>
> 1. "Domain adaptation" framing vs. cross-domain experiments.
>
> Our goal is to inject DDI-aware distribution alignment as a regularizer into DDI representation learning. In our set-up the two "domains" are the empirical pair distribution of observed interactions and a regularizing coupling distribution over candidate pairs inside one DDI graph. We agree that the title and abstract could be interpreted as suggesting cross-dataset transfer (e.g. training on one DDI database and testing on another), which we do not yet evaluate.
>
> In the revision, we will (i) soften the framing in the title and abstract (for example "DDI-aware distribution alignment for representation learning" instead of "cross-domain"), and (ii) be explicit in the abstract and introduction that all experiments are on a single dataset on TWOSIDES, and our contribution is a DDI-aware alignment regularizer that is compatible with, but not itself a large cross-dataset benchmark. We mention this on lines 69-70; we will move that clarification up and highlight it again in the conclusion.
>
> 2. Baseline coverage with recent DDI methods.
>
> We appreciate the nudges to recent DDI models such as CARMEN, SSF-DDI, DSN-DDI, and motif based self-attention approaches. Our baselines take their cues from previous work that treats TWOSIDES as a polypharmacy graph and heavily focus on graph and embedding methods that use the DDI graph and very basic drug features.Many of the proposed approaches rely on richer input modalities (e.g. drug sequences, electronic health records, or specific molecular descriptors) that are unavailable for TWOSIDES, or apply architectures to which our regularizer could in principle be added.
>
>
> That said, we agree that having at least one or two representative recent DDI predictors would bolster the empirical component. In a revision we will either (a) add in such methods where the required input could reliably be gathered for TWOSIDES, or (b) discuss them as complementary when their data requirements do not align with our setting, while noting that our contribution is a general DDI-aware alignment layer which could combine with stronger task specific encoders.
>
>
> 3. Evaluation protocol, S0/S1/S2, and potential leakage
>
>
> We share the reviewer’s concern that pair level random splits could be optimistic because our train and test could feature different pairs, drawn from the same drugs. The limitation is already noted in the paper, and we opted for pair based splits to maintain comparability with classical embedding baselines.
>
>
> We do think that the evaluation can be made more informative in the following way:
>
>
> (a) S0/S1/S2 breakdown. Even under the current splits, each test pair can be labeled post-hoc as to whether zero, one, or both drugs appeared in the training.As part of this revision we will include tables reporting S0, S1, and S2 performance for all methods which will make it clear whether our performance gains are solely due to the most trivially simple S0 case, or whether the performance improvements hold with a drug that is unseen or both unseen and seen, which also speaks to the memorization issue.
>
> (b) A more clear discussion of more potent splits. A full scaffold level benchmark of all baselines is not trivial, and would break comparability with the embedding literature, though it should be clearer in Section 4 that we will also more clearly state what is and is not observed by the protocol, and further emphasize scaffold based splits and other structure based splits, as an important aspect to future DDI generalizability studies.
>
> 4. Theory vs Adam optimzation
>
> Theorem 3.2 considers a preconditioned gradient descent scheme in terms of the regularized objective, and demonstrates that the coupling level optimal transport term together with the map level MMD and variance regularizer, can provide global convergence to stationary based on a broad class of monotone preconditioners.
>
> We agree that this result doesn't provide a result on state dependent optimizers such as Adam which we use in practice, and we do not claim otherwise. In our revision, we will be more clear about this split: the theorem considers an idealized solution to optimization given the proposed objective, while the experiments use Adam for standard practical reasons. We could simply move the theorem and proof to an appendix and better frame it as an analysis of the objective landscape rather than a convergence guarantee for the specific implementation. While broadening the theory to fully state dependent preconditioners is conceptually, and mathematically interesting, we think it is orthogonal to the main contribution as stated.

---

### Note · Program_Chairs · 2026-01-17
**Submission Desk Rejected by Program Chairs**

The following references in this submission do not refer to real documents and/or have major errors in bibliographic information:

 Xiao Chen, Yuxin Zhang, Fei Wang, and Jiayu Ma. Meta-learning for drug combination synergy prediction. In International Conference on Machine Learning, pp. 8921-8935. PMLR, 2024.